# Probiotics and Their Bioproducts: A Promising Approach for Targeting Methicillin-Resistant *Staphylococcus aureus* and Vancomycin-Resistant *Enterococcus*

**DOI:** 10.3390/microorganisms11102393

**Published:** 2023-09-25

**Authors:** Manav Jain, Gideon Stitt, Luke Son, Elena Y. Enioutina

**Affiliations:** Division of Clinical Pharmacology, Department of Pediatrics, Spencer Fox Eccles School of Medicine, University of Utah, Salt Lake City, UT 84108, USA; u6052750@utah.edu (M.J.); gideon.stitt@hsc.utah.edu (G.S.); luke.son@hsc.utah.edu (L.S.)

**Keywords:** multi-drug resistant bacteria, methicillin-resistant *Staphylococcus aureus* (MRSA), vancomycin-resistant *Enterococcus* (VRE), conventional treatment, alternative treatment, probiotics, antimicrobial peptides, bacteriocins

## Abstract

Antibiotic resistance is a serious global health problem that poses a threat to the successful treatment of various bacterial infections, especially those caused by methicillin-resistant *Staphylococcus aureus* (MRSA) and vancomycin-resistant *Enterococcus* (VRE). Conventional treatment of MRSA and VRE infections is challenging and often requires alternative or combination therapies that may have limited efficacy, higher costs, and/or more adverse effects. Therefore, there is an urgent need to find new strategies to combat antibiotic-resistant bacteria. Probiotics and antimicrobial peptides (AMPs) are two promising approaches that have shown potential benefits in various diseases. Probiotics are live microorganisms that confer health benefits to the host when administered in adequate amounts. AMPs, usually produced with probiotic bacteria, are short amino acid sequences that have broad-spectrum activity against bacteria, fungi, viruses, and parasites. Both probiotics and AMPs can modulate the host immune system, inhibit the growth and adhesion of pathogens, disrupt biofilms, and enhance intestinal barrier function. In this paper, we review the current knowledge on the role of probiotics and AMPs in targeting multi-drug-resistant bacteria, with a focus on MRSA and VRE. In addition, we discuss future directions for the clinical use of probiotics.

## 1. Introduction

Multi-drug resistant (MDR) bacterial infections represent a significant threat, with an estimated 1.3 million attributable deaths worldwide in 2019 [1]. According to the Centers for Disease Control and Prevention (CDC) Antibiotic Resistant Threats in the United States Report, anti-microbial-resistant pathogens were responsible for over 2.8 million infections and 35,000 deaths annually from 2012 through 2017 [2]. The European Center for Disease and Control (EDCD) also estimates that nearly 70% of the 600,000 cases of antibiotic-resistant bacterial infections and 27,000 attributable deaths in Europe were caused by MDR gram-negative bacteria in 2015 [3]. Additionally, cross-transmission of MDR bacteria between pets, livestock, and humans is an increasing concern for veterinarians and healthcare providers [4]. Intimate contact with companion animals (e.g., dogs and cats) carrying MDR bacteria, including methicillin-resistant *Staphylococcus aureus* (MRSA), vancomycin-resistant enterococci (VRE), and carbapenemase-producing Enterobacterales, may represent a direct threat to their owners. Several reports of MRSA transmission from animals to humans have been noted in the literature [5].

While strategies on how to best treat these infections using conventional antimicrobial therapy continue to evolve, significant challenges remain. Among these challenges are substantial toxicity of many conventional antimicrobial drugs, altered pharmacokinetics (PK) in critically ill patients, the lack of standardized and universally available therapeutic drug monitoring (TDM) for many first-line and “last resort” antimicrobial drugs, and the increasing number of extensively resistant and pan-drug-resistant strains due to poor antimicrobial stewardship practices. Recent guideline updates by the Infectious Diseases Society of America (IDSA) and the European Society of Clinical Microbiology and Infectious Diseases (ESCMID) attempt to address this growing threat [6,7,8,9]. Still, innovative strategies are necessary to effectively address this challenge.

One potential alternative to conventional antimicrobial treatment of MDR infections could be the use of probiotics or their bioproducts. According to the World Health Organization, probiotics are “live microorganisms which, when administered in adequate amounts, confer a health benefit on the host” [10]. Probiotics became popular among consumers promoting healthy lifestyles. They are commonly used as functional foods or dietary supplements in the US and European Union countries and are offered in supermarkets, pharmacies, and online. Available probiotics include, but are not limited to, *Lactobacillus rhamnosus* G.G., *Lactobacillus reuteri*, *Lactobacillus casei*, *Lactobacillus paracasei*, *Bacillus coagulans*, *Bacillus clausii*, *Bifidobacterium infantis*, *Bifidobacterium longum*, *Bifidobacterium infantis, Streptococcus thermophilus*, and yeasts, including *Saccharomyces boulardii* and *Saccharomyces cerevisiae*. Many probiotics are blends of two or more different species [11]. The US Food and Drug Administration (FDA) regulates probiotic safety and health claims, and many probiotics are awarded a Generally Recognized as Safe (GRAS) status [12].

There are strong supporting data for probiotic use to prevent or treat the following five distinct disorders: necrotizing enterocolitis, acute infectious diarrhea, antibiotic-associated diarrhea, acute respiratory tract infections, and infant colic [11,13]. Antimutagenic [14,15], anticarcinogenic [16], and antidiarrheal properties [16], immune modulation [17], prevention of atopic dermatitis [18], and the ability to lower blood cholesterol levels [19] are a few of the other known activities of probiotics. With an established basis of utilizing probiotics for infectious diseases, they represent a potential tool against emerging resistant infections due to their ability to interfere with pathogen growth, quorum sensing, and biofilm formation [20,21,22].

Probiotic bacteria also produce bioactive constituents able to exert antimicrobial properties (e.g., exopolysaccharides, enzymes, cell wall fragments, antimicrobial peptides, and several other bioactive molecules) [23]. Antimicrobial peptides (AMPs) are of particular interest among these constituents. Approximately 380 AMPs of bacterial origin have been discovered, and seven have been approved for clinical use [24]. AMPs of bacterial origin are referred to as bacteriocins. A common classification of bacteriocins separates compounds into several classes, each of which may be further subdivided based on shared functional or structural similarities [24,25]. AMPs are low molecular weight proteins produced with various microorganisms, including Gram-positive and Gram-negative bacteria, bacteriophages, viruses, and fungi [26,27]. All naturally occurring bacteriocins are translation products of the bacterial ribosome. The antibacterial mechanisms of AMPs comprise of the direct killing of microorganisms and modulation of the host immune response [26]. Multiple clinical trials have been conducted to establish the efficacy of AMPs in human infections, including phase I, II, and III trials (see [26] for review).

This review article aims to summarize the current knowledge on the role of probiotics and AMPs in targeting MDR bacteria, with a focus on MRSA and VRE, and discusses future directions for their clinical use.

## 2. Conventional Treatment of Multi-Drug Resistant Infections

Conventional treatment recommendations for MDR bacteria are generally broken down by the site of infection and causative pathogen [6]. Treatment recommendations are subsequently put forward based on specific infection sites and pathogen combinations (Table 1). While describing the recommended treatment for each specific situation falls outside the scope of this work, general trends can be drawn from the guidelines put forth by the North American and European infectious diseases societies. 

Extended-spectrum β-lactamase-producing Enterobacterales (ESBL-E) represent a group of Gram-negative bacteria of growing importance, with a 53% increase in identification in bacterial cultures in the United States from 2012 to 2017 [28]. ESBL enzymes are able to inactivate most β-lactam antibiotics, such as penicillins, cephalosporins, and aztreonams. However, carbapenems and non-β-lactam antibiotics remain active against ESBL-E organisms (Table 1 and [6]). Recommended agents for the treatment of ESBL-E range from nitrofurantoin, trimethoprim-sulfamethoxazole (TMP-SMX), ciprofloxacin, levofloxacin, or a carbapenem for uncomplicated cystitis to preferred treatment with a carbapenem for ESBL-E infections outside of the urinary tract [6,9]. In the case of extended-spectrum β-lactamase-producing organisms or AmpC-producing organisms, cefepime is the agent recommended against bacteria at moderate to high risk of AmpC enzyme production. Non-β-lactam options for invasive infections include TMP-SMX or a fluoroquinolone [6].

Over 13,000 infections and 1000 deaths occur annually in the United States due to carbapenem-resistant Enterobacterales (CRE) infections [2]. Treatments options for urinary tract infections due to CRE include nitrofurantoin for uncomplicated cystitis, and TMP-SMX, ciprofloxacin, levofloxacin, ceftazidime-avibactam, meropenem-vaborbactam, Imipenem–cilastatin–relebactam, and cefiderocol for uncomplicated cystitis, pyelonephritis, or complicated urinary tract infection (cUTI). For non-urinary tract infections due to CRE, depending on the specific β-lactamase present, meropenem-vaborbactam, ceftazidime-avibactam, Imipenem–cilastatin–relebactam, or cefiderocol are the preferred agents [6,9].

MDR *Pseudomonas aeruginosa* represents a significant threat, with 32,600 infections and 2700 deaths in the United States in 2017 (Table 1 and [6]). When susceptible to fluoroquinolones or traditional β-lactams such as Piperacillin–tazobactam and cefepime, these agents are preferred over carbapenems. For isolates resistant to traditional β-lactams, Ceftolozane–tazobactam, ceftazidime-avibactam, Imipenem–cilastatin–relebactam, and cefiderocol are the preferred options [6,9].

Carbapenem-resistant *Acinetobacter baumannii* (CRAB) is a uniquely challenging organism, as these organisms are frequently isolated from respiratory samples or wounds, and there is no true “standard of care” against which to compare treatment regimens (Table 1 and [6]). High-dose ampicillin–sulbactam is the backbone of therapy against CRAB, with combination therapy recommended where it is possible to utilize minocycline, tigecycline, polymyxin B, or cefiderocol [6,9]. 

Finally, *Stenotrophomonas maltophilia*, an organism ubiquitous in aquatic environments, offers a therapeutic challenge due to difficulty in identifying colonization versus a pathogenic isolate, multiple antimicrobial resistance genes, and lack of “standard of care” (Table 1 and [6]). Recommended treatment consists of either the combination of ceftazidime–avibactam and aztreonam, or any two of the agents TMP-SMX, minocycline, tigecycline, cefiderocol, or levofloxacin [6].

### 2.1. Conventional Treatment of Methicillin-Resistant Staphylococcus aureus and Vancomycin-Resistant Enterococcal Infections

Two of the most challenging to treat Gram-positive bacterial organisms are MRSA and VRE. While the number of MRSA infections has decreased over time, there were still 323,700 cases in hospitalized patients and 10,600 attributable deaths in 2017 [2]. Treatment of MRSA infections is highly dependent on whether the patient is hospitalized or not, as well as the site of infection. Simple abscesses and boils can often be treated with incision and drainage alone, whereas antibiotic therapy is needed for skin and soft-tissue infections (SSTI), and oral antibiotic options include clindamycin, TMP-SMX, doxycycline, minocycline, or linezolid. For more serious SSTI in hospitalized patients, intravenous (IV) vancomycin, daptomycin, telavancin, and either oral or IV clindamycin or linezolid are all reasonable options [8]. For MRSA bacteremia and infective endocarditis, vancomycin or daptomycin are recommended, with linezolid and clindamycin replacing daptomycin for the treatment of MRSA pneumonia [8].

Similar to MRSA, absolute numbers of VRE infections have decreased over time, but 54,500 cases in hospitalized patients and 5400 deaths still occurred in 2017 [2]. A number of agents are active against VRE, including daptomycin, linezolid, tedizolid, teicoplanin, dalbavancin, telavancin, oritavancin, and several tetracycline derivatives [29]. For the treatment of infective endocarditis secondary to VRE, the American Heart Association recommends the use of linezolid or daptomycin. In the case of persistent bacteremia due to VRE endocarditis, combination therapy with daptomycin plus ampicillin or ceftaroline may be utilized [30]. Though less common, VRE may also be the cause of osteoarticular infections, particularly in people with prosthetic joints. Based on adequate demonstrated bone penetration, daptomycin may be used to treat isolates with lower minimal inhibitory concentrations (MICs). Additionally, daptomycin shows activity against VRE biofilms, particularly when used in combination with ampicillin, ceftaroline, ertapenem, or rifampicin [31]. Linezolid also holds the potential for treating these infections; however, hematologic toxicities may be a limiting factor due to the necessary prolonged duration of therapy [29].

### 2.2. Challenges of Conventional Treatments of Multi-Drug Resistant Infection

The challenges of treating MDR infections are numerous. Difficulty determining antibiotic susceptibility in the microbiology laboratory is an issue for both Gram-negative and Gram-positive organisms. Rapid identification of specific β-lactamases may be difficult and complicates antibiotic choice for serious Gram-negative infections [6]. Meanwhile, VRE suffers from reproducibility issues between broth microdilution and E-test methods for determination of isolated strain sensitivity to antibiotics [29]. Many antibiotics discussed here carry significant toxicities ranging from acute kidney injury to myelosuppression when used at high doses for extended durations [7,29]. PK parameters are frequently altered in critically ill patients relative to healthy volunteers, resulting in a wide range of plasma drug exposures for a given dose [32]. TDM is available to guide the dosing of many agents, but implementation is not universal or standardized, and optimal target exposures are not well-defined for all antibiotics of interest [33]. The recent COVID-19 pandemic resulted in a loss of progress in the fight against MDR organisms, with reported rates increasing in 2020 [34]. Finally, the utility of currently available antimicrobial agents is waning as antibiotic resistance continues to grow. Patients presenting with extensively drug-resistant or pan-drug-resistant isolates is a clinical reality, leaving clinicians no choice but to select the “least bad” treatment option rather than an agent optimized for the bacterial isolate at hand [35]. Considering the myriad of challenges facing healthcare workers in the fight against MDR infections, innovative strategies are crucial to counter this threat.

## 3. Probiotics and Their Bioactive Compounds Interfering with the Growth of Pathogenic Microorganisms

Several studies describe the antimicrobial activity of various probiotic strains, AMPs, and other antimicrobial molecules produced with these strains capable of inhibiting bacterial growth, disrupting biofilm formation, or interfering with quorum sensing, with the majority of studies utilizing in vitro models. For example, Lahtinen et al. reported that 3 out of 38 *Bifidobacterium* strains isolated from elderly adults inhibited the growth of *S. aureus* [36]. Hydrogen peroxide and heat-stable proteinaceous compounds produced with these strains were responsible for antimicrobial activity. In another report, the antibacterial activity of *Lactobacillus casei* IMV B-7280, *L. acidophilus* IMV B-7279, *Bifidobacterium longum* VK1, and *Bifidobacterium bifidum* VK2 was evaluated in an animal model of intravaginal staphylococcosis [37]. While all strains were capable of suppressing growth of *S. aureus* individually, the combination of probiotics demonstrated more vigorous antimicrobial activity. It appeared that the antimicrobial activity of these strains was associated with their ability to improve host immune system activity. Regularly consuming *Bacillus* probiotics may also decrease *S. aureus* colonization [38]. Piewngam et al. demonstrated that *Bacillus* species produce lipopeptides which inhibit quorum sensing with *S. aureus*. Finally, the probiotic *Pediococcus acidilactici* HW01 reduced the production of the virulence factors with *Pseudomonas aeruginosa* (e.g., protease, pyocyanin, and rhamnolipid) [39].

Due to growing antibiotic resistance, *Helicobacter pylori* is a difficult-to-treat infection [40]. Probiotics have been successfully used in conjunction with conventional antibiotic regimens to eradicate *H. pylori* and reduce the chance of developing antibiotic resistance strains [40]. It has been reported that heat-killed lactobacilli inhibited the *H. pylori* growth; however, exact mechanisms of growth inhibition are unknown [41].

According to a study by Lagrafeuille et al., the supernatant from *Lactobacillus plantarum* CIRM653 significantly impaired *Klebsiella pneumoniae* biofilm formation [42]. The supernatant interfered with the activity of biofilm-related genes and downregulated operons relevant to quorum detection. Additionally, *Bifidobacterium longum* 5(1A) reduced *K. pneumoniae* infection in mice by producing proinflammatory cytokines, increasing neutrophil recruitment, and decreasing bacterial load [43]. The *Bifidobacterium* treatment reduced the mortality rates of mice infected with *K. pneumoniae* from 50% to 0%.

Candida species’ capacity to form biofilms on both biotic and abiotic surfaces determines how harmful they can be. The formation of fungal biofilms may increase resistance to anti-fungal drugs and protect the fungus from immune cells. The crude extracts of 13 *Lactobacillus* strains had anti-*Candida* action, with MICs ranging from 1.25 to 10 mg/mL [44]. The bacteriocin produced with *Lactobacillus fermentum* SD11 demonstrated antibacterial activity against Gram-positive and Gram-negative bacteria as well as yeast [45]. Crude filtrate supernatants from *Lactobacillus acidophilus*, *Lactobacillus rhamnosus*, *L. plantarum*, and *Lactobacillus reuteri* had an impact on the development of *Candida albicans* hyphae and biofilm formation [46]. Additionally, recent studies showed that *Lactobacillus* species can decrease the development of *C. albicans* hyphae by secreting antimicrobial substances, including biosurfactants [47,48]. In vitro studies have demonstrated that *L. rhamnosus* GR-1 and *L. reuteri* RC-14 prevent biofilm formation with *Candida glabrata* via downregulation of biofilm-related genes EPA6 and YAK1 [49]. A scanning electron microscopy analysis showed minimal biofilm formation in mixed cultures of probiotic lactobacilli and *C. glabrata*. In another study, *Lactobacillus salivarius* decreased biofilm formation and the number of colonies of *Streptococcus* mutants and *C. albicans* in mixed cultures [50]. The authors concluded that *L. salivarius* may secrete bioactive molecules responsible for its antimicrobial activity.

Some studies suggest that probiotics can successfully treat infections by affecting host immune cells, and probiotic strains often exert their anti-fungal effects via regulation of immune responses. Rossoni et al. found that *L. paracasei* 28.4 prolonged the survival of *Galleria mellonella* larvae, a honeybee parasite, infected with *C. albicans* by activating the immune system of the larvae and increasing the number of hemocytes [51]. Interestingly, *L. paracasei* 28.4 was also able to upregulate genes responsible for the production of anti-fungal peptides. Another group of researchers constructed a *L. casei* strain which secreted bovine lactoferrin (BLF) [52]. The *L. casei* pPG612.1-BLF strain decreased the infection burden in mice with vaginal candidiasis, with resultant reductions in local interleukin-17 (IL-17) concentrations. In this case, the decreased production of IL-17 could be due to the decreased number of fungi present in the vagina of infected mice or due to the ability of lactoferrin to reduce IL-17 production with activated immune cells. It has been shown that *Streptococcus salivarius* inhibits *Propionibacterium acnes* using secreting bioactive molecules [53]. *S. salivarius* K12 decreased production of interleukin-8 (IL-8) with epithelial cells and keratinocytes, and *Enterococcus faecalis* administration resulted in a 50% reduction of inflammatory lesions in patients with acne [53]. 

Rosignoli et al. tested whether topical application of *Lactobacillus johnsonii* could inhibit *S. aureus* adhesion to the skin and increase innate skin immunity [54]. The results showed that the application of the probiotic suspension reduced *S. aureus* adhesion by up to 74% and modulated endogenous expression of AMPs. A mixture of oral probiotics significantly decreased the scoring atopic dermatitis (SCORAD) index, a clinical tool assessing dermatitis severity, in the treatment group and reduced topical steroid use in moderate atopic dermatitis patients [55]. Topical probiotics have also been shown to improve atopic and seborrheic dermatitis by increasing skin ceramides, improving erythema, flaking, and itching, and reducing *S. aureus* microbial load. The most commonly used probiotics include *S. thermophiles*, *V. filiformis*, *S. hominis*, *S. epidermidies*, and *L. johnsonii* [56,57,58].

### 3.1. Probiotics and Their Bioactive Compounds Interfering with the Growth of MRSA

The antimicrobial activity of *L. acidophilus* and *L. casei* against clinical isolates of MRSA was demonstrated by Karska-Wysocki and colleagues using the agar diffusion method [59]. The growth inhibition of MRSA with *L. acidophilus* ranged from 1.7 to 2.9 cm, and ranged from 1.4 to 2.9 cm with *L. casei*. The authors suggested that growth inhibition was possibly due to lactic acid production with probiotic strains. Purified *L. lactis* biosurfactants demonstrated antibacterial efficacy against diverse strains of *S. aureus*, including MRSA [60]. *L. acidophilus*, *L. casei*, and *L. plantarum* additionally demonstrated in vitro synergistic activity against MDR staphylococci, including several clinical MRSA isolates [61]. The antibacterial activity of *L. paracasei* BMK2005, a strain isolated from healthy infants’ feces, was examined by Bendajeddou et al. [62]. The researchers discovered that *L. paracasei* BMK2005 had antibacterial activity against MRSA clinical isolates and Gram-negative bacteria resistant to cefotaxime and ceftazidime (e.g., *Escherichia coli*, *Klebsiella oxytoca*, *Enterobacter cloacae*, and *P. aeruginosa*). It appeared that growth suppression in these bacteria was associated with the ability of *L. paracasei* to produce paracaseicin A, a heat-stable bacteriocin. Chen et al. observed that different MRSA strains could not grow in the presence of the supernatants produced with *Lactobacillus fermentum*, *Bifidobacterium longum*, and *Bifidobacterium animalis* subsp. lactis [63]. These results emphasize the remarkable potential of the probiotic-derived bioactive molecules capable of inhibiting the proliferation of MRSA.

*S. aureus*, including MRSA, can form biofilms [64]. The ability of these strains to form biofilms can significantly increase the mortality and morbidity of MRSA-infected patients. The anti-biofilm activity of various probiotics has been demonstrated by several researchers. Walencha et al. reported that biosurfactant production with *L. acidophilus* reduced biofilm production with MRSA strains in vitro [65]. Lactic acid produced with strains of *L. plantarum* K.F. and *L. casei* Y1 were tested in another study for their ability to inhibit MRSA S547 growth and biofilm formation [66]. Both *Lactobacillus* strains could reduce the growth of MRSA in the agar well diffusion assay and demonstrated a time-dependent effect on biofilm formation with MRSA S547. These data may support the use of biosurfactants as bio-detergents to remove staphylococcal biofilms from biological habitats, surgical instruments, catheters, and more.

A combination of two bacteriocins, micrococcin P1 and garvicin KS, showed synergistic activity against biofilms generated in vitro with *S. aureus* and MRSA strains (Table 2) [67]. The highly resistant MRSA strain’s sensitivity to the antibiotic penicillin G was also restored with this bacteriocin-based antibacterial combination.

Quorum sensing interference via probiotics has recently received much attention [68]. It has been demonstrated, for example, that lactic acid-producing bacteria produced bioactive molecules such as bacteriocins, hydrogen peroxide, and organic acids which inhibit bacterial quorum sensing. *L. acidophilus* accomplished this via suppression of autoinducer 2 production [68]. Autoinducer genes were responsible for bacterial communication when the bacterial density reached a particular limit. Zhang et al. showed that *B. subtilis* CFS also significantly downregulated the expression of genes responsible for quorum sensing, biofilm formation, and ability of *S. aureus* to adhere to tissues [69]. Additionally, *B. subtilis* inhibited the mecA gene, a gene responsible for β-lactam antibiotic resistance, and interfered with penicillin-binding protein 2a production resulting in increased antibiotic susceptibility. Abbasi et al. tested the ability of synbiotics, synergistic combinations of probiotics and prebiotics, to modulate the expression of mecA gene expression in staphylococci [70]. When the antibacterial activity was tested in a well diffusion assay, the supernatants produced with synbiotics suppressed the growth of six clinical MRSA isolates. Co-culture of supernatants with MRSA decreased mecA expression and increased susceptibility to antibiotics.

It has been reported that *L. fermentum* is able to interfere with MRSA adhesion to Caco-2 cell lines [71]. *L. fermentum* demonstrated superior adherence to Caco-2 cells compared to MRSA, and a substantial decrease in MRSA adherence was seen when Caco-2 cells were pre-incubated with *L. fermentum*. After pre-infecting Caco-2 cells with MRSA for two hours and adding *L. fermentum* thereafter, adherent MRSA was successfully displaced from the Caco-2 cells. Earlier, it was found that *L. fermentum* produced a bacteriocin that was active against MRSA [72].

AMPs isolated from fungi may represent another practical therapeutic approach for the treatment of MDR infections. Micasin, a defensin-like peptide isolated from *Microsporum canis*, demonstrated inhibitory activity against Gram-positive and Gram-negative bacteria, including MRSA (Table 2) [73]. In one study, 79% of mice pre-treated with micasin and intraperitoneally infected with a lethal dose of MRSA P1386 survived the challenge. It was determined that micasin exerts its antibacterial activity through interference with intracellular protein folding in bacteria. Additional examples of the effectiveness of probiotic strains producing AMPs can be found in Table 2.

**Table 2 microorganisms-11-02393-t002:** Probiotic strains producing AMPs against MRSA.

Species/Strain	AMP	Activity against	Reference
*Lactococcus garvieae*	Garvicin KS	MRSA (along with micrococcin P1), *Bacillus*, *Listeria*, *Enterococcus*, along with polymyxin B or nisin, acts against *A. baumannii* and *E. coli*	[67,74,75]
*Enterococcus faecalis* 28 and 93	Enterocin DD28 and DD93	MRSA	[76]
*L. plantarum*	Plantaricin NC8 β Synthetic Linopeptide	MRSA Linopeptide acts against both Gram-positive and negative bacteria	[77]
*M. canis*	Micasin	MRSA, *P. aeruginosa*	[73]
*Streptomyces actinomycinicus* PJ85	Actinomycin D and DGLA AMP	MRSA	[78]
*Bacillus* spp. T12	Bacin A1, A2, A3, A4	MRSA	[79]
*Enterococcus faecalis* 14	Enterocin DD14	Along with methicillin, acts against MRSA	[80]
*L. lactis*	Nisin Z	Along with methicillin, acts against MRSA	[81]
*Staphylococcus capitis*	Nisin J	MRSA	[82]
*Bacillus* spp.	Mersacidin	MRSA	[83]
*Lactobacillus pentosus*	Pentosin JL1	MRSA	[84]
*Bacillus subtilis*	BAC-IB17	MRSA	[85]
*Enterococcus durans*	Durancin 61A	MRSA, *C. difficile*	[86]
*Bacillus subtilis* subsp. *spizizenii*	Ba-49	MRSA	[87]
*Bacillus* spp. MMA	Subtilomycin	*L. monocytogenes*, *Clostridioides* sporogenes, MRSA, and heterogeneous vancomycin-intermediate-level-resistant *S. aureus*. Also active against some Gram-negative bacteria, including *Aeromonas hydrophila*, *Vibrio anguillarum, Alteromonas* spp., and *P. aeruginosa*	[88,89]
*Bacillus licheniformis*	Lichenicidin	*B. subtilis*, *B. pumilus*, *Bacillus megaterium*, *S. aureus* (methicillin-sensitive and methicillin-resistant strains)	[89]
*B. amyloliquefaciens* GA1	Amylolysin	MRSA, *Bacillus* spp. *Enterococcus* spp., *Listeria monocytogenes*	[89]
*Staphylococcus epidermidis strain* 224	Epidermicin NI01	MRSA	[90]

### 3.2. Probiotics and Their Bioactive Compounds Interfering with the Growth of VRE

VRE is a bacterium that is difficult to treat with conventional methods. The treatment of VRE-infected patients with probiotics or AMPs could be an alternative to the conventional antibiotic approach. Two bacteriocin-like peptides produced with *Vagococcus fluvialis* were found to possess strong antibacterial activity against Gram-positive bacteria, including MDR *Enterococcus faecium*, when used in combination (Table 3) [91]. The mechanism by which these peptides exert their effect is via the creation of pores in the bacterial cell wall, resulting in cell death.

Another study demonstrated that the lactic acid-producing bacterium *L. lactis* MM19 produced a bacteriocin called nisin. Daily intragastric treatment of mice colonized with VRE with nisin reduced fecal levels of VRE by up to 2.5 times [92]. *Lactobacillus murinus* Y74 and *L. plantarum* HT121, when given to VRE-infected mice, significantly reduced VRE colonization in the intestines and restored the diversity of the intestinal microbiota [93].

*E. faecalis* EF478 produced a bacteriocin-like molecule that effectively inhibited the growth of MDR enterococci, including VRE (Table 3) [94]. The results of the analysis show that this bacteriocin is serine protease. The bacteriocin also showed excellent chemical and thermodynamic characteristics, suggesting it might be stable in an in vivo environment. 

Enterocin K1, a leaderless bacteriocin produced with *L. lactis*, was highly active against nosocomial MDR *E. faecium* (Table 3) [95]. Enterocin K1 is attached to the membrane-bound protein RseP and forms pores that allow the leakage of solutes and other cellular components. Since RseP is essential for the enterococcal bacterial stress response, a bacteriocin that targeted RseP was capable of killing both sensitive and resistant bacteria [95]. *L. lactis* KA-FF 1-4 produced antimicrobial substances that were effective in suppressing the growth of VRE in vitro [96]. After 6 h of co-culturing *L. lactis* KA-FF 1-4 and VRE, the number of VRE CFU/mL dropped to zero from the initial 10^3^–10^4^ CFU/mL. However, its anti-VRE activity was decreased to only 3.59–6.12% in the human colon model, potentially due to the interfering activity of human gut microbiota. In a separate study, extracellular vesicles, membrane-containing vesicles produced with *L. plantarum* WCFS1, protected *C. elegans* from VRE challenge with the activation of host defenses [97]. 

Combining conventional antimicrobial treatment with probiotics or bacteriocins produced with the probiotics may result in a synergistic effect and increase the effectiveness of treatment. Bucheli et al. investigated the combined effects of antibiotics and bacteriocins from *Enterococcus* and *Pediococcus* against VRE [98]. Combining bacteriocins with vancomycin was found to increase the antibacterial effect of vancomycin. At the highest concentrations of bacteriocin and vancomycin, VRE viability inhibition increased up to 90%. The addition of nisin and vancomycin to VRE culture results in a remarkable reduction in the MIC for vancomycin [99]. Synergistic results were also observed when nisin was added to VRE incubated with either ampicillin or chloramphenicol.

Sun et al. identified a cocktail of probiotics consisting of *Bacillus coagulans*, *L. rhamnosus* GG, *L. reuteri,* and *L. acidophilus* using microbial network analysis [100]. This study found that all four strains, when administered individually, showed weak or no efficacy against VRE, while the combination of these probiotics drastically lowered the population of VRE and inhibited VRE adherence to Caco-2 cells by downregulating multiple VRE host-adhesion genes. Possible additional mechanisms include regulation of quorum sensing mechanisms in enterococcus, direct competitive inhibition with *L. rhamnosus* GG due to presence of a pilus gene cluster, and bacteriocin production with Lactobacillus.

**Table 3 microorganisms-11-02393-t003:** Probiotic strains producing AMPs against VRE.

Species/Strain	AMP/Antimicrobial Substance	Activity against	Reference
*Vagococcus fluvialis*	Vagococcocin T	Broad Gram-positive coverage except *S. aureus*	[91]
*Pediococcus acidilactici* MM33	Pediocin PA-1/AcH	*Enterococcus* sp. *Lactobacilli*, *Listeria*	[92,101]
*E. faecalis*	EF478	VRE	[94]
*L. lactis*	*Enterococcus* K1	VRE	[95]

### 3.3. Probiotics and Their Bioproducts Interfering with the Growth of MRSA and VRE

Cacaoidin, the lanthidin produced with *Streptomyces cacaoi*, showed antimicrobial activity against VRE and MRSA by inhibiting peptidoglycan biosynthesis (Table 4) [102]. A unique peptide, lactomodulin, a peptide produced with *L. casei* subsp. *rhamnosus*, was effective against MRSA with an MIC_50_ of 0.2 umol and VRE at MIC_50_ of 0.4 umol (Table 4) [103]. Lactomodulin also reduced the production of inflammatory cytokines (e.g., IL-8, IL-6, IL-1β, and TNF-α) by the activated Caco-2 cell line without a significant cytotoxic effect. Garvicin KS (a multi-peptide bacteriocin produced with a strain of *L. garvieae*) had wide antimicrobial activity against Gram-positive infections, including MRSA and VRE, as well as the food-borne diseases caused by *Listeria* and *Bacillus* spp. *Streptomyces antibioticus* strain M7 (isolated from the rhizospheric soil of *Stevia rebudiana*), which indicated that it also had potent antibacterial activity against various pathogenic bacteria, including MRSA, VRE, and Gram-negative bacteria (Table 4) [104]. The strain produced antibacterial compounds identified as actinomycins V, X2, and D. Actinomycin X2 was more effective than D and V. Aunpad et al. isolated a broad spectrum bacteriocin, pumilicin 4, from *Bacillus pumilis* WAPB4 effective against MRSA and VRE (Table 4) [105].

The antimicrobial peptide plectasin, isolated from the saprophytic ascomycete *Pseudoplectania nigrella*, was found to have low potency against MRSA (MIC 16 → 128 μg/mL) and VRE (MIC 32 → 128 μg/mL) [106]; however, its variant (NZ2114), when tested with teicoplanin, had reduced MIC values against VRE (MIC 16 μg/mL) (Table 4) [107]. The molecular target of plectasin NZ2114 appears to be the bacterial cell wall precursor lipid II. Plectasin NZ2114 also showed dose-dependent efficacy in a model of MRSA infective endocarditis [108]. Additional examples of antimicrobial activity of AMPs against MRSA and VRE are in Table 4.

**Table 4 microorganisms-11-02393-t004:** Probiotic strains producing AMPs against MRSA and VRE.

Species/Strain	AMP/Antimicrobial Substance	Activity against	Reference
*Streptomyces cacaoi*	Cacaoidin	MRSA, VRE, *C. difficile*	[102]
*L. casei*	Lactomodulin	MRSA, VRE, *C. difficile*, *B. cereus*, *L. monocytogenes*, *E. coli*	[103]
*Streptomyces antibioticus* M7	Actinomycin V, X2, D	MRSA. VRE, *B. subtilis*, *K. pneumoniae*, *S. epidermidis*, *S. typhi*	[104]
*Bacillus pumilus*	Pumilicin 4	MRSA, VRE, *B. subtilis*, *B. licheniformis*, *B. thurigenesis*	[105]
*Psuedoplectania nigerella*	Plectasin	MRSA, VRE, *S. pneumoniae*, *C. diphtherium*, *C. jeikeium*	[106]
*Streptomyces* spp.	Neoactinomycin A and B	MRSA, VRE	[109]
*L. lactis*	Nisin Z (along with lacticin 3145)	*E. faecium*, MRSA, *Lactobacillus* sp., Gram-negative bacteria	[92,101,110]
*Enterococcus faecalis* 11	KT11	MRSA, VRE. Broad antimicrobial spectrum	[111]
*Scedosporium apiospermum*	Scedosporisin	MRSA, VRE	[112]
*Bacillus amyloliquefaciens*	YD1	MRSA, VRE	[113]
*B. subtilis* subsp. spizizenii DSM 15029	Entianin	MRSA, VRE	[89]
*Bacillus cereus* as-1.1846	Cerecidins	MRSA, VRE	[114]

## 4. Clinical Evidence for the Effectiveness of Probiotics for the Treatment of MRSA Infection

The results of clinical trials assessing the effectiveness of probiotics for eradiation of the MRSA infection are presented in Table 5. There is a shortage of published clinical data on the use of probiotics in the prevention or treatment of clinical MRSA infections. In one case report, after multiple courses of antibiotic therapy, an elderly patient who had undergone intra-abdominal surgery experienced recurrent diarrhea caused by MRSA [115]. Additionally, MRSA was isolated from anterior nares. The treatment of the patient with oral vancomycin, *Saccharomyces boulardii*, and topical mupirocin 2% resulted in the resolution of diarrhea and negative MRSA cultures in the stool and nares. While the combined treatment with conventional antibiotics and a yeast probiotic was beneficial for the patient, this regimen remains controversial. In a prospective clinical study, 50 adult patients undergoing living-donor liver transplantation were randomly assigned to a group receiving synbiotic therapy (*B. breve*, *L. casei*, and prebiotic galactooligosaccharides) for 2 days before surgery followed by 2 weeks after surgery or a control group [116]. The synbiotic group had only one incident of systemic infection (4%) compared to the control group, 24% of which had various infections caused by MRSA and *Enterococcus* spp. The authors concluded that the complications due to infections significantly decreased in liver transplantation patients after symbiotic therapy.

Probiotic lactobacilli were found to be helpful in eliminating persistent MRSA carriage in the nose and throat according to a study conducted by Roos et al. [117]. Seven patients with documented upper airway MRSA carriage for at least a 1-year period, negative MRSA cultures from other body sites, failure of previous decolonization attempts, and an absence of skin lesions were enrolled. Each nostril received a self-administered dose of four different *Lactobacillus* strains every morning. At bedtime, each nostril received an oral suspension which was ingested. Treatment continued until the first MRSA cultures were negative. After 3 to 7 months of treatment, 5 throat MRSA carriers had negative throat cultures. Three of these five participants also carried MRSA in the nares, all of which were cleared. All three of the participants cleared their nasal cultures before their throat cultures. These participants remained culture-negative for 10 to 37 months following therapy. Only two additional participants who were throat carriers continued to test consistently positive for MRSA after nearly a year of treatment.

Two clinical studies investigated whether the treatment with *L. rhamnosus* could reduce MRSA colonization in nasal, oropharyngeal, intestinal, and axillary sites [118,119]. The first study demonstrated a reduction in nasal and intestinal colonization by 33% and 50%, respectively. The second study demonstrated a 73% reduction in the odds of MRSA presence, and an 83% reduction in the odds of any *S. aureus* present in stool samples. 

An additional study demonstrated that *Bacillus subtilis* probiotic administration resulted in a 96.8% reduction in *S. aureus* present in stool and 65.4% reduction in the nose [120]. The authors suggested that the use of this probiotic may decrease MRSA prevalence in patients.

**Table 5 microorganisms-11-02393-t005:** Summary of clinical studies related to probiotic use against MRSA.

Study Type	Study Population	Intervention	Key Findings	Reference
Case report	Adult	Oral vancomycin 250 mg twice daily for 10 days, *Saccharomyces boulardi* 250 mg twice daily for 15 days, topical mupirocin 2% for 5 days.	Negative MRSA culture after 11 days and 5 months of therapy.	[115]
Prospective randomized control study	Adult	Fifty adult patients undergoing live-donor liver transplant between September 2005 and June 2009 randomized into a group receiving 2 days of preoperative and 2 weeks of postoperative synbiotic therapy (*Bifidobacterium breve, Lactobacillus casei*, and galactooligosaccharides (synbiotic group) and a group without synbiotic therapy (control group).	4% vs. 25% post-transplant infection in synbiotic vs. control group.	[116]
Prospective study	Adult	7 patients given 4 species of *Lactobacillus* via oral and intranasal routs (5 × 10^9^ CFU/mL). Treatment given till first negative MRSA culture.	5 patients had throat clearance, 3 of which also had nasal clearance. Culture negative from 10–37 months after therapy.	[117]
Randomized double-blind, placebo-controlled trial	Adult	48 subjects randomized either to probiotic group (*L. rhamnosus HN001* 10^10^ CFU/capsule, one capsule per day) for 4 weeks or control.	At the four-week time point, MRSA was detected in 67% and 50% of probiotic group subjects with nares and the gastrointestinal tract colonization, respectively. Three subjects who initially tested positive for VRE were negative after four weeks of probiotic exposure.	[118]
Randomized double-blind, placebo-controlled trial	Adults	113 subjects once daily oral probiotic or placebo capsule administered for four weeks. The probiotic capsule contained 1 × 10^10^ colony forming units of *L. rhamnosus HN001*.	Odds of MRSA presence in probiotic vs. placebo group was 0.27 (0.07–0.98) in any sample and 0.17 (0.04–0.73) in stool samples.	[119]
Randomized double-blind, placebo-controlled trial	Adults	115 participants randomly assigned and received 250 mg of probiotic B subtilis MB40 or placebo once per day for 30 days.	Oral administration of probiotic *B. subtilis* resulted in significant reduction of *S. aureus* in the stool (96.8%; *p* < 0.0001) and the nose (65.4%; *p* = 0.0002), whereas there were no significant differences in the placebo groups.	[120]

### Clinical Evidence for the Effectiveness of Probiotics for the Eradication of VRE Colonization

The results of clinical trials of the effectiveness of probiotics for eradiation of VRE are presented in Table 6. One small clinical study investigated the efficacy of *L. rhamnosus* G.G. against intestinal VRE colonization of participants [121]. Participants received 100 g of yogurt containing either probiotic or no probiotic for four consecutive weeks. Fecal samples were evaluated for the presence of VRE during the treatment and 8 weeks after treatment completion. Only 1 of the 12 patients in the placebo group had cleared their VRE, whereas all 11 patients in the probiotic group were VRE negative after treatment completion. Eight probiotic patients were still VRE negative at the 8-week follow-up, while three patients in the probiotic group who previously received antibiotics became VRE positive again. Eight of the 11 control participants who initially remained VRE positive were crossed over to probiotic treatment, and all subsequently cleared their VRE. The investigators concluded that although larger clinical trials are necessary, *L. rhamnosus* G.G. can be utilized to eradicate VRE.

Cheng et al., in a case series, reported VRE eradication in four patients using a series of procedures, including the use of polyethylene glycol to wash out VRE and treatment with linezolid or daptomycin followed by treatment with *L. rhamnosus* G.G. [122]. All patients tested negative for VRE and remained negative for 23–137 days after treatment completion. In the reported cases, it is unclear what role *L. rhamnosus* G.G. played in the irradiation of VRE or whether the addition of probiotics to the regimen was responsible for the long-term clearance of bacteria.

Buyukeren et al. investigated how *L. rhamnosus* G.G. therapy affects the elimination of VRE in the gastrointestinal tract of colonized newborns [123]. The probiotic group received a daily dose of one million CFU of *L. rhamnosus* G.G. until three negative cultures were achieved or up to 6 months. Twenty-one out of 22 patients were found to be free of VRE within 6 months of treatment.

Of note, other studies have demonstrated weak or no activity of *L. rhamnosus* GG in VRE eradication. Szachta et al. investigated the ability of *L. rhamnosus* G.G. to eradicate VRE from the intestines of 65 children [124]. The treatment had only a temporary effect with a substantial reduction of VRE colonization after three weeks of intervention in the probiotic group compared to the control group. Unfortunately, the difference between the probiotic and control groups disappeared by week 4 after treatment discontinuation. Eight adult patients treated with *L. rhamnosus* Lcr35, a strain similar to *L. rhamnosus* G.G. strain, did not exhibit a visible decline in the density of VRE colonization [125]. A 2-week intervention in an adult population with 20 billion CFU of *L. rhamnosus* GG per day also did not affect VRE colonization [126]. No difference in the VRE count was observed between treatment and control group at any time point. This could be attributed to the shorter intervention and low *L. rhamnosus* GG dosage. Rauseo et al. conducted a study to ascertain the impact of *L. rhamnosus* G.G supplementation on antibiotic-resistant microorganisms, including VRE, in hospitalized patients taking antibiotics [127]. The authors concluded that *L. rhamnosus* G.G. administration neither accelerated the reduction of antibiotic-resistant microorganism colonization (*L. rhamnosus* G.G. 18% vs. placebo 24%; OR, 1.44; 95% CI, 0.27–7.68) nor slowed down antibiotic-resistant organism acquisition. Another study sought to determine if supplementation with *Lactobacillus rhamnosus* G.G. for 4 weeks had a beneficial effect on hospitalized adults carrying vancomycin-resistant *E. faecium* (VREfm) [128]. The number of patients who had cleared VREfm colonization following the intervention was the primary outcome. After the 4-week intervention, there was no impact caused by *L. rhamnosus* G.G. on the elimination of VREfm. The inclusion of a geriatric population, more extended hospitalization, and treatment with vancomycin could be some of the reasons for the failure to clear VREfm. Regt et al. used multiple species of probiotic powder and found no role for the cocktail of probiotics in the prevention of VRE colonization [129]. However, the probiotic-strain-specific response of VRE isolates to treatments remains to be evaluated. A microbial network analysis-selected probiotic mixture [100] was used in a clinical trial aimed to identify an effective probiotic cocktail capable of competing with VRE in human gut microflora (ClinicalTrials ID: NCT03822819). However, the results of the trial were not published. In a separate retrospective analysis, it was found that a persistent VRE outbreak in a hospital for early rehabilitation was prevented by the treatment of patients with *S. boulardii* and *E. coli* Nissle [130].

These studies show mixed results, indicating that VRE eradication with only probiotics may be difficult to achieve. The addition of synergistic probiotics or bioproducts secreted with probiotics could help to overcome this problem. The careful selection of probiotic strains or their bioproducts and optimization of regimens are necessary.

**Table 6 microorganisms-11-02393-t006:** Summary of clinical studies related to probiotics use against VRE.

Study Type	Study Population	Intervention	Key Findings	Reference
Double-blind, randomized, placebo-controlled trial	Adult	Twenty-seven patients randomly assigned to either a treatment group (receiving 100 g daily of yoghurt containing *L. rhamnosus* G.G. for 4 weeks) or a control group (receiving standard pasteurized yoghurt).	A total of 11/11 treatment participants cleared VRE and 8 remained VRE negative after 4 weeks of trial completion, 1/12 control participants cleared VRE. After cross over to *L. rhamnosus* G.G. containing yogurt group, all controls cleared VRE.	[121]
Case series	Adult	A total of 4 patients, 2 with end-stage liver disease, 1 with post-liver transplant infection, and 1 with complicated infective endocarditis were given bowel preparation, linezolid, and daptomycin followed by *L. rhamnosus* G.G. (80 mg).	Three out of four showed VRE clearance.	[122]
Open-label prospective follow-up	Newborn	Forty-five patients were randomly assigned to either a treatment group receiving *L. rhamnosus* G.G. 10^6^ CFU or control group until three consecutive negative cultures or for maximum 6 months.	A total of 21/22 in treatment group vs. 12/23 patients in control group achieved eradication at the end of therapy.	[123]
Randomized, single blind	Children	Sixty-one patients were randomly assigned to either a treatment group receiving *L. rhamnosus* G.G. 3 billion CFU or placebo for 21 days with no follow-up until the end of therapy.	A total of 20 out of 32 patients in treatment group vs. 7 out of 29 in control group were VRE negative at the end of treatment.	[124]
Randomized, double blind	Adult	Nine patients were randomly assigned to either a treatment group receiving *L. rhamnosus* Lcr35 for 5 weeks (10^9^ CFU) or placebo with no follow-up until the end of therapy.	VRE negative culture achieved in 2/6 patients in treatment group vs. ½ patients in placebo group.	[125]
Double-blind, randomized, placebo control	Adult	*L. rhamnosus* G.G., 10^10^ CFU twice daily for 14 days vs. placebo with 7 days of a follow-up period.	No difference in VRE counts at any time point. No difference in CFU in subjects who received *L. rhamnosus* GG.	[126]
Double-blind, randomized, placebo control	Adult	*L. rhamnosus* G.G. 10^10^ CFU vs. placebo until hospital discharge.	Odds of antibiotic resistance organism (ARO) acquisition for VRE in treatment vs. placebo 1.32. The odds of ARO loss for VRE in treatment vs. placebo 0.31 (0.02–4.41).	[127]
Double-blind, randomized, placebo control	Adult	*L. rhamnosus* G.G. 60 × 10^9^ CFU vs. placebo.	A total of 12/21 vs. 15/27 patients cleared VRE in the treatment vs. placebo group.	[128]
Prospective cohort with cross-over	Adult	*L. rhamnosus* G.G. 10^9^ CFU was given to patients with >48 h of hospital stay twice daily until discharge.	Probiotics did not prevent VRE acquisition.	[129]

## 5. What Is Next?

Probiotics have been extensively studied due to their popularity among consumers. Probiotics appear to be effective against antibiotic-induced *Clostridioides difficile*-associated, traveler’s, and viral diarrheas [131]. The effectiveness of probiotics for these conditions has been confirmed in several clinical trials [131]. Probiotics could also be effective for prophylaxis and the treatment of atopic dermatitis, Crohn’s disease, rheumatoid arthritis, colon cancer, and high cholesterol levels.

However, Suez et al. stressed that even high-quality clinical trials may “point toward opposing conclusions and remain conflicting” [132]. Several clinical trials reported the effectiveness of probiotics in treating infections in children, adults, and elderly patients. At the same time, other studies claimed that probiotics did not affect the outcomes of their studies. The authors believed that these discrepancies may have been due to differences in the outcomes measured by different studies, for example, the patient’s well-being, which is entirely subjective vs. changes in the measurable values (e.g., cytokine levels or C-reactive protein). Another contributing factor to the discrepancies in reported results could be the type of probiotic strains used.

Our review of available publications investigating the efficacy of probiotics against MDR bacteria revealed that in vitro and animal model studies demonstrated that probiotics could improve mucus barrier integrity, efficiently suppress the growth of MDR bacteria, prevent biofilm formation, interfere with quorum sensing, and modulate host immune responses against pathogens [20,21,22]. However, the specific mechanisms of probiotic antibacterial activity are not well elucidated. Rubio et al. proposed that extracellular vesicles released by probiotics may be responsible for their antibacterial action [133]. Extracellular vesicles are produced with many bacteria and are involved in cell–cell communications [134]. The content of the extracellular vesicles is diverse and depends upon what strain of bacteria produces the vesicles [133]. The constituents found in vesicles may inhibit pathogenic bacteria and stimulate innate and adaptive immune defenses (e.g., cytokine and immunoglobulin production) [133].

Numerous preclinical studies demonstrated the antibacterial activity of probiotics, but only randomized clinical trials can support health claims [12]. Often, probiotics are used as a “last resort” in patients with MDR infections. Some clinical data suggest that probiotics are effective as a monotherapy or adjunct for the treatment of infectious diseases [12].

The main concerns with probiotic use for treatment of diseases, such as probiotic strain characteristics, quality of probiotic products, regimen of probiotic administration, and safety in vulnerable patient populations, were discussed at the 2022 International Scientific Association for Probiotics and Prebiotics (ISAPP) [135]. The strain-related issues included the effects on MDR organisms, presence of genes encoding toxins, and antibiotic resistance in probiotic strains [135]. A potential danger could be the transfer of antibiotic-resistant plasmids from probiotics to antibiotic-sensitive pathogens [136,137]. Concerns regarding probiotic quality comprised the use of good manufacturing practices for probiotic production, finding a safe and efficacious dose for patients of various age, and ensuring potency throughout the shelf life of the product [135]. The safety issues are one of the main concerns with probiotic use in patients, especially in patients with impaired immune functions (Figure 1). Severely ill patients may have dysregulated immune defenses due to underlying infection or the use of immunosuppressive drugs (e.g., in organ transplant patients) [138,139,140]. Immune system changes associated with aging also lead to a predisposition to autoimmune and oncological diseases and increased incidence of bacterial and viral infections [141]. It is not clear whether the use of live probiotics will lead to systemic infection, “probiotiemia”, and excessive stimulation of the already activated immune system (Figure 1 and [12]). Several cases of systemic probiotic infections and endocarditis have been reported following the use of *Lactobacillus*, *Streptococcus*, and *Bacillus* strains [12]. Patients with effectively functioning immune systems are likely able to effectively fight systemic infections secondary to probiotics; however, patients with suppressed immune function will require additional interventions to control the probiotic infection. It has also been reported that the use of probiotics resulted in a long-term gastrointestinal colonization caused by the probiotics. Probiotics could potentially affect the structure and functions of the patient’s microbiome and disrupt the gut epithelial barrier (Figure 1 and [135]).

Therefore, we feel that AMPs produced with the probiotic strains could be a good alternative to live probiotics in patients with altered immune function (Figure 1). Lantibiotics, class I bacteriocin AMPs synthesized with Gram-positive bacteria, are small heat-stable peptides [26]. Lantibiotics effectively inhibit Gram-positive bacteria and food-borne pathogens [142]. Lantibiotics usually demonstrate antibacterial activity against closely related strains of bacteria. Nisin, mentioned in this review, inhibits cell wall synthesis by forming a complex with lipid II [143]. The lantibiotic, mersacidin, exerts its antimicrobial activity via induction of the cell wall stress response [144].

Non-lanthionine-containing heat-stable class II bacteriocins are typically small molecules capable of destabilizing bacterial cell membranes, pore formation, and cell death [25,26,142]. Class III AMPs (e.g., lysostaphin, enterolysin A), unlike classes I and II, are heat-labile and possess enzymatic endopeptidase activities, resulting in bacterial cell wall disruption [26,142]. Gram-negative bacteria also produce AMPs named colicins or colicin-like microcins [26]. These AMPs may target ribosomal RNA or transfer RNA or degrade DNA [145].

One of the advantages of using AMPs is that microorganisms cannot develop a resistance to them; therefore, they may represent an important tool in the treatment of MDR bacterial infections [26]. Unfortunately, there are some drawbacks for the immediate use of natural AMPs in clinical practice as they are susceptible to proteolytic degradation and have low oral bioavailability (Figure 1 and [146]). 

The treatment of infections caused by MDR organisms, including MRSA and VRE, presents multiple challenges to healthcare professionals. Among them are the implementations of standardized TDM practices and definition of optimal target exposures for all antibiotics of interest. There is a limited selection of conventional antibiotics to treat the growing number of MDR and pan-drug-resistant bacteria. Therefore, the exploration of new and safer treatments for these infections is critical. One solution could be the use of probiotics and their bioproducts. There are sufficient preclinical data demonstrating that probiotics and their bioproducts are effective against MDR organisms, including MRSA and VRE. As research continues, new probiotic strains are being discovered and are demonstrating their efficacy in treating MDR infections, but the lack of translation of this research to clinical practice limits the wide use of probiotics. The main concerns with probiotic use in patients are the standardization of the manufacturing process, use of optimal regimens, and safety in immunocompromised and fragile patients. We believe that extensive randomized clinical trials are needed to evaluate the efficacy and safety of probiotics and AMPs in patients with infectious diseases and different immune statuses. There is a need to improve the composition of AMPs to make them more resistant to degradation and more suitable for oral route administration. While the use of probiotics and their bioproducts are not free from challenges, they have the potential to be used as an alternative to conventional antibiotics or as adjunct therapy against MDR organisms, including MRSA and VRE (Figure 1). 

## Figures and Tables

**Figure 1 microorganisms-11-02393-f001:**
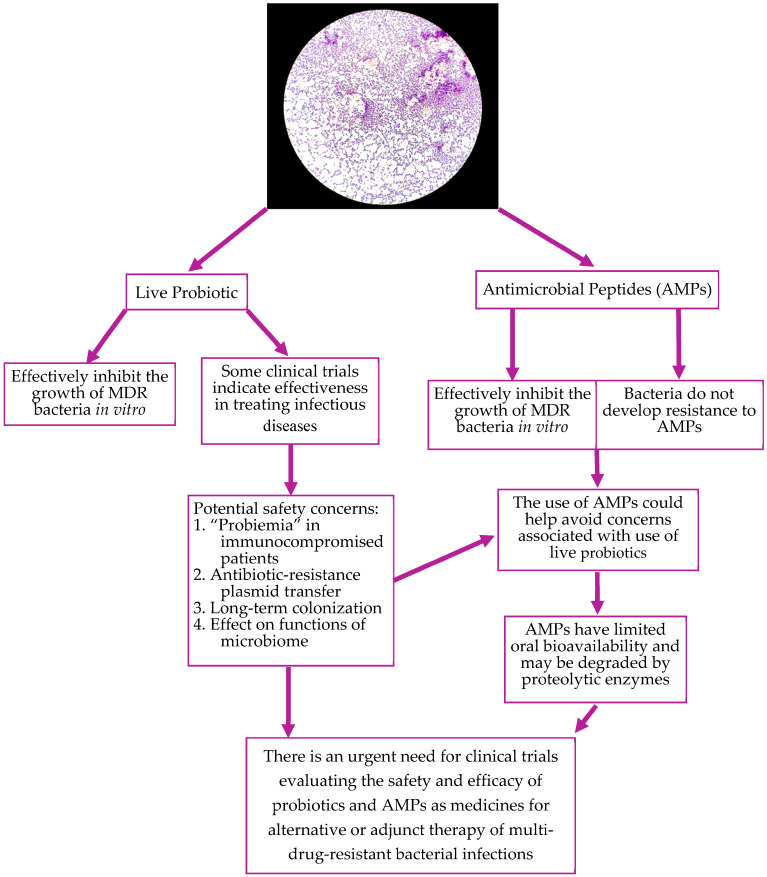
Pros and cons of probiotics and AMP use for the treatment of MDR bacterial infections.

**Table 1 microorganisms-11-02393-t001:** Conventional treatment options for multi-drug resistant bacterial infections [6,9].

Organism	Clinical Scenario	Conventional Treatment Options
Extended-spectrum β-lactamase-producing Enterobacterales (ESBL-E)	Uncomplicated cystitis	Nitrofurantoin TMP-SMX * Ciprofloxacin Levofloxacin Carbapenem Single-dose aminoglycoside Fosfomycin (*E. coli* only)
	Non-urinary tract	Meropenem Imipenem–cilastatin Ertapenem TMP-SMX (oral step-down) Ciprofloxacin (oral step-down) Levofloxacin (oral step-down)
	Carbapenem-resistant ESBL-E	Ceftazidime–avibactam Meropenem–vaborbactam Imipenem–cilastatin–relebactam Cefiderocol
Amp-C β-lactamase-producing Enterobacterales	Cefepime MIC ** < 4 µg/mL	Cefepime
	Cefepime MIC ≥ 4 µg/mL	Carbapenem (with demonstrated activity)
Carbapenem-resistant Enterobacterales (CRE)	Uncomplicated cystitis	Nitrofurantoin TMP-SMX Ciprofloxacin Levofloxacin Ceftazidime–avibactam Meropenem–vaborbactam Imipenem–cilastatin–relebactam
	Pyelonephritis/Complicated urinary tract infection	Nitrofurantoin TMP-SMX Ciprofloxacin Levofloxacin Ceftazidime–avibactam Meropenem–vaborbactam Imipenem–cilastatin–relebactam
	Non-urinary tract	Meropenem–vaborbactam Ceftazidime–avibactam Imipenem–cilastatin–relebactam Cefiderocol
Multi-drug-resistant *Pseudomonas* *aeruginosa*	Traditional β-lactam susceptible	Fluoroquinolone Piperacillin–tazobactam Cefepime
	Traditional β-lactam resistant	Ceftolozane–tazobactam Ceftazidime–avibactam Imipenem–cilastatin–relebactam Cefiderocol
Carbapenem-resistant *A. baumannii* (CRAB)	All	High-dose ampicillin–sulbactam PLUS one of the following: Minocycline Tigecycline Polymyxin B Cefiderocol
*Stenotrophomonas maltophilia*	All	Ceftazidime–avibactam PLUS aztreonam OR any two of the following: TMP-SMX Minocycline Tigecycline Cefiderocol Levofloxacin

* TMP-SMX: trimethoprim-sulfamethoxazole; ** MIC: minimum inhibitory concentration.

## Data Availability

Not applicable.

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
