# Peer review of "Probiotics and Their Bioproducts: A Promising Approach for Targeting Methicillin-Resistant Staphylococcus aureus and Vancomycin-Resistant Enterococcus"

_microorganisms, 2023, doi:10.3390/microorganisms11102393_

Round 1

Reviewer 1 Report

The comments are inserted as sticky notes in the attached PDF.

Minor editing of the English language required

Author Response

Thank you for your very helpful feedback.

Below, please find our response to the reviewer’s comments and suggestions.

  1. please specify the title to only MRSA and VRE since the review discussed these only.

Answer: While the review is also talking about other pathogenic species causing infectious diseases in humans, the reviewer is correct that the main focus of the manuscript is MRSA and VRE.  We edited the title per the reviewer’s request.

  1. The authors should mention the spreading of antibiotic-resistant bacteria in different sources in the environment as animals and its product

Answer: It is a very good suggestion. We have added to the Introduction sentences that domesticated and companion animals can be a source of MDR bacteria for humans. Please see lines 34-40.

  1. Section 2.0. The reviewer suggested potentially using two references.

Answer: Dear reviewer,   the references that you have suggested are about the antimicrobial properties of medicinal plants and another paper about gas reduction in pigs by using several strains of probiotics.

We decided not to use these articles for one reason: the focus of the manuscript under review is the treatment of MDR bacterial infection in humans using probiotics. The senior author of this manuscript is an expert in the anti-microbial/immunomodulatory properties of medicinal plants. I will use this reference in the upcoming webinar. 

  1. The authors should refer to challenge of using the probiotics and bioproducts.

Answer: It was briefly mentioned in association with the use of probiotics in immunocompromised patients and the instability and poor oral bioavailability of bioproducts. Per the review’s request, we expanded this section in the  Discussion. Please see lines 434-442 and 450-452.

  1. Please rewrite the conclusion to be more specific and related to the study

Answer: The summary has been rewritten.

Reviewer 2 Report

In the review article "Probiotics and their Bioproducts: A Promising Approach for Targeting Antibiotic-Resistant Bacteria", the authors presented in great detail, with a critical review, the potential of probiotics and their metabolites in the fight against infections caused by MDR. I really liked the way the authors connected the powerlessness of conventional therapy against MDR with the potential of probiotic bacteria.

In the entire paper, the Latin names of probiotic bacteria should be revised, as well as the rule when writing the full name of the bacteria (at the first appearance in the text) and the use of abbreviations in the following text.

Author Response

Thank you for your suggestion. Our colleague Ashley Rogers, who is board-certified in Medical Microbiology,  reviewed the paper for correct nomenclature and microorganism abbreviations.   

Round 2

Reviewer 1 Report

The authors responded to commey